# Penetration Enhancers for Topical Drug Delivery to the Ocular Posterior Segment—A Systematic Review

**DOI:** 10.3390/pharmaceutics13020276

**Published:** 2021-02-18

**Authors:** Abhinav Thareja, Helen Hughes, Carmen Alvarez-Lorenzo, Jenni J. Hakkarainen, Zubair Ahmed

**Affiliations:** 1Institute of Inflammation and Ageing, College of Medical and Dental Sciences, University of Birmingham, Birmingham B15 2TT, UK; a.thareja@bham.ac.uk; 2Department of Science, Waterford Institute of Technology, Cork Road, Waterford X91 K0EK, Ireland; hhughes@wit.ie; 3Departamento de Farmacología, Farmacia y Tecnología Farmacéutica, I+DFarma, Facultad de Farmacia and Health Research Institute of Santiago de Compostela (IDIS), Universidade de Santiago de Compostela, 15782 Santiago de Compostela, Spain; carmen.alvarez.lorenzo@usc.es; 4Experimentica Ltd., Microkatu 1, P.O. Box 1199, 70211 Kuopio, Finland; jenni.hakkarainen@experimentica.com

**Keywords:** topical drug delivery, penetration enhancers, neuroprotection, retina, retinal ganglion cells, posterior segment

## Abstract

There is an unmet clinical need for eye drop formulations to efficiently treat the diseases of the posterior ocular segment by non-invasive topical administration. Here, we systematically reviewed the literature on ocular penetration enhancers and their ability to transfer drugs to the posterior segment of the eye in experimental studies. Our aim was to assess which penetration enhancer is the most efficient at delivering drugs to the posterior segment of the eye, when topically applied. We conducted a comprehensive search in three electronic databases (Ovid Embase, Ovid MEDLINE, and PubMed) to identify all the relevant manuscripts reported on ocular penetration enhancers based on the PRISMA guidelines. We identified 6540 records from our primary database search and filtered them per our inclusion/exclusion criteria to select a final list of 14 articles for qualitative synthesis. Of these, 11 studies used cell penetrating peptides (CPPs), 2 used chitosan, and 1 used benzalkonium chloride (BAC) as the penetration enhancer. Cationic and amphipathic CPPs, transactivator of transcription (TAT), and penetratin can be inferred to be the best among all the identified penetration enhancers for drug delivery to the fundus oculi via topical eye drop instillation. Further high-quality experimental studies are required to ascertain their quantitative efficacy.

## 1. Introduction

Eye diseases originating in the posterior segment ocular tissues, mainly in the retina-choroid, are a major cause of visual impairment across the globe. It is estimated that worldwide, around 2.2 billion people have some sort of visual impairment and out of these, people affected by diseases of the ocular posterior segment including age-related macular degeneration (AMD) (196 million) and diabetic eye disease (146 million) constitute a large proportion [1]. Specifically, in Europe, AMD and diabetic eye disease lead to a critically high incidence of blindness in older populations and together constitute approximately 35% of the causes of blindness, thus putting a huge burden on public health expenditure [2]. AMD is the reason for 8.7% blindness globally with projected estimates at 196 million people having this disease in 2020 and expanding to 288 million by 2040 and is also the single and most widely recognized cause of elderly blindness in developed nations, especially in individuals over 60 years of age [1,2,3]. Studies also provide strong evidence that AMD is more prevalent in people of European ancestry as compared to other ethnicities, with Europe having the highest prevalence of AMD anywhere in the world [2,3,4]. In the working adult population of the world and particularly in Europe however, diabetic eye disease comprising diabetic retinopathy (DR) and diabetic macular edema (DME) is the most common reason for vision impairment and blindness, with a prevalence of approximately 34.6% among global diabetic population and 29.4% in diabetic Europeans, and an estimated increase from current 6.4 million affected people in Europe to 8.6 million in 2050 [5,6].

Topical delivery of drug formulations through the eye surface, mainly via eye drops, to treat diseases of the posterior segment is significantly inhibited by the various physiological and anatomical defensive ocular barriers including clearance by lacrimal fluid and drainage mechanisms, conjunctival systemic absorption, cornea, and different tissues in the anterior segment. On the other hand, systemic administration of drugs is hindered by the blood–ocular barriers, comprising blood–aqueous barrier (BAB) and blood–retina barrier (BRB), resulting in ocular bioavailability of less than 5% [7,8]. The only feasible conventional routes of drug administration to the posterior segment are intravitreal and periocular injections that are highly invasive, cause huge patient discomfort, thus low compliance and require specialist clinicians, besides leading to serious complications including endophthalmitis, increased intraocular pressure, cataract, glaucoma and even retinal detachment in rare cases [9,10,11]. Another option is sustained-release drug-eluting ocular implants, but that too requires surgical intervention, suffers from common post-operative complications, and needs to be replaced after a time.

Among various emerging technologies to optimize topical ocular drug delivery congruent with an increased drug bioavailability towards the posterior segment tissues, use of penetration enhancers (PEs) could be the most promising and viable means to achieve drug formulations that exhibit much higher uptake, cellular internalization, and longer retention into the posterior segment via simple eye drop instillation. As the name suggests, ocular PEs are chemical or biological agents capable of modifying the various defensive barriers and membranes of cells and tissues in the eye, through known and unknown mechanisms, to affect an enhanced passage across them. Multiple types of PEs have been used in pharmaceutical formulations and reported in literature, such as cyclodextrins (CD), calcium chelators (EDTA), surfactants (BAC), bile acids and bile salts, fatty acids, azones, mucoadhesive polymers (chitosan and HA), and cell-penetrating peptides (CPP) [12,13,14]. PEs can be used as additives or excipients; as drug carriers conjugated with the drug; or as conjugates with drug carriers like micro/nanoparticles, liposomes, micro/nanoemulsions, etc. The majority of these PEs barring a few, however, are effective only for transcorneal permeation to enhance bioavailability in the anterior ocular segment and have negligible or no effect on uptake in the tissues of the ocular posterior segment.

There are a few literature reviews on different reported PEs for topical ocular drug delivery, but none specifically on the PEs that can access the ocular posterior segment. The present report systematically reviewed the available experimental literature on PEs that can efficiently enter the posterior segment of the eye after topical administration with the aim of identifying the most effective PE for non-invasive delivery of drugs to the posterior segment of the eye. This will help in developing non-invasive, novel pharmaceutical formulations with enhanced drug bioavailability in the ocular posterior segment to treat the complex eye diseases like AMD and diabetic eye disease through easy to apply, non-invasive and cost-effective mode of eye drop instillation.

## 2. Materials and Methods

### 2.1. Review Process

This systematic review was conducted based on the guidelines outlined by Preferred Reporting Items for Systematic Reviews and Meta-Analyses (PRISMA) [15,16]. The reporting parameters were modified in the context of animal studies. All of the review steps were duly performed by two authors (A.T. and Z.A.) independently and any conflicts or disagreements were resolved after discussion to reach a consensus.

### 2.2. Literature Search

Published reports on PEs were identified by running exhaustive searches in three electronic databases: Ovid Embase, Ovid MEDLINE, and PubMed from the time of inception to 30 December 2020. We formulated a common search string for all the databases containing the terms ((penetration enhancers) AND (retina OR retinal ganglion cells OR retinal pigment epithelium)). The search was restricted to English language reports only. Duplicates were removed and manuscripts were initially shortlisted by referencing the titles and abstracts. Final selection was done by reviewing full texts against the inclusion and exclusion criteria by the two authors (A.T. and Z.A.). Additional publications were identified upon reviewing the references of shortlisted articles.

### 2.3. Inclusion and Exclusion Criteria

To be eligible for review, the screened articles had to fulfil certain criteria: (a) only manuscripts of completed experimental studies published in a journal were included; conference abstracts and in-process or non-indexed citations were not included; (b) only studies that investigated topical route of administration of PEs were included; (c) only animal studies were included; clinical or human trials or randomised controlled trials (RCTs) were not; (d) rat, mouse, and rabbit models only for live animal testing; (e) most importantly, only those studies were included where PEs were able to reach the tissues of the ocular posterior segment mainly retina-choroid in-vivo, and studies must demonstrate and compare in-vivo biodistribution and efficacy in the posterior segment of eyes with and without PEs, by standard techniques like immunohistochemistry assays and/or microscopy. Eligibility assessment of studies was performed in an unblinded standardized manner.

### 2.4. Risk of Publication Bias

Risk of bias (RoB) was evaluated using a modified version [17] of SYRCLE’s risk of bias tool for animal studies [18], which itself is adapted from the Cochrane Collaboration’s tool for assessing risk of bias in RCTs [19]. Briefly, each article was assessed on 13 different risk parameters and the responses were recorded as: (a) yes, if the parameter was positively reported in the article with sufficient information on its methodology; (b) yes, if the parameter was reported positively but with insufficient information or without a methodology; or (c) no, if the parameter was negatively reported or not reported at all. In all of the final shortlisted studies, we surveyed the various critical aspects of bias that play a vital role in animal experiments, to determine the methodological quality of the included studies.

### 2.5. Data Extraction

Data was collected and summarized in a predesigned electronic data extraction form with multiple sections to address all of the key aspects of the study including author(s), journal and year of publication, type of PE used, in-vivo animal model, and efficacy and biodistribution in the posterior segment tissues in-vivo. Since efficacy of drug penetration to the posterior chamber was mostly analysed by immunohistochemistry, meta-analysis was not possible and hence we qualitatively reviewed the reports to answer our review aims.

## 3. Results

### 3.1. Study Selection

We retrieved 6540 records from our primary literature search on the databases and after filtering duplicates, 5559 unique original published articles were identified. These were then subjected to a preliminary screening of titles and abstracts, and a list of 28 articles was found to be relevant for review. The whole focus was on identifying PEs that can efficiently access or facilitate penetration of cargo to the fundus oculi by topical eye drop instillation, and to the best of our knowledge we obtained all-inclusive search results that correspond to our eligibility criteria. Full texts of these 28 shortlisted manuscripts were then examined, any relevant references were also analysed and a final selection of 14 eligible studies was made for qualitative synthesis in our review after application of inclusion and exclusion criteria. The literature identification strategy is demonstrated by means of a PRISMA flow diagram in Figure 1.

Of the 14 selected studies, 11 or around 78.6% were exclusively dealing with CPPs as PEs for delivery to the ocular posterior segment and the rest (*n* = 3 or 21.4%) were based on other compounds. Thirteen studies (92.9%) employed rat or mouse models for in-vivo efficacy and biodistribution tests and one study (7.1%) used rabbits. Since none of the studies quantitatively analysed penetration efficiency to the posterior segment, a meta-analysis of the data was not possible and hence what follows is a qualitative analysis of the results presented for the different PEs. The characteristics of the included studies are defined in Table 1.

CPPs can be defined as short chain peptides, often consisting of 30 or less amino acid residues, that exhibit a well-defined property of internalization across cellular membranes with negligible cytotoxicity, through an energy dependent and/or independent mechanism, without the need of chiral interactions with surface receptors. The most commonly studied CPPs are of cationic nature, however a number of anionic or hydrophobic peptides have also been reported. CPPs are able to transport a multitude of covalently or non-covalently linked cargoes of different molecular weights into living cells [34]. Different CPPs have been reported for drug delivery across the skin topically, nasal delivery for pulmonary diseases, and even for crossing the blood–brain barrier (BBB), though the mechanism of their cellular internalization is still debated and is a contentious topic [34,35]. Lately, there has been a rigorous use of CPPs for ocular drug delivery, mostly to the anterior segment tissues of eye and here we report the use of CPPs as penetration enhancers for drug delivery to the ocular posterior segment. Of the total 11 CPP candidates in our review, 5 reports were on transactivator of transcription (TAT), 4 on penetratin, and 2 on other peptides.

#### 3.1.1. TAT

TAT is a protein transduction domain (PTD) with general sequence GRKKRRQRRRPQ of the HIV Tat protein and is one of the most highly investigated CPP for drug delivery. Wang et al. (2010) [21] first demonstrated its potential to deliver a therapeutic growth factor across the posterior segment tissues by eye drop instillation. They investigated the penetration ability of TAT conjugated human acidic fibroblast growth factor (aFGF) in the retina and its therapeutic efficiency in-vivo in a retinal ischemia–reperfusion (IR) injury model in 18 week-old male Sprague–Dawley rats. A primary immunohistochemical analysis was carried out to assess tissue penetration of TAT-aFGF compared with non-conjugated aFGF, where TAT-aFGF was readily detected at 30 min post administration in the retinas whereas non-modified aFGF was not detected. TAT-aFGF was localized mainly in retinal ganglion cells (RGCs) in the ganglion cell layer (GCL), its concentration peaked around 30 min to 1 h, and gradually declined thereafter, but could still be detected up to 8 h post administration.

These results were reinforced with the histological examination of retinal injury from IR after application of TAT-aFGF and aFGF where retinas from TAT-aFGF treated eyes showed significantly less severe pathological changes, lesser damage to tissue morphology, and lower depletion in RGCs than the untreated groups, whereas aFGF only treated groups were no different from untreated eyes. Furthermore, TUNEL staining of diseased retinas affirmed a lower instance of RGC apoptosis induced by IR upon treatment with TAT-aFGF. Apoptotic RGCs in the GCL of TAT-aFGF treated retinas were significantly less in numbers than in the groups treated with aFGF only. In addition, TAT-aFGF treated groups showed accelerated recovery of retinal function compared to those treated with aFGF as measured by electroretinography. The authors suggested a non-corneal route of transport to the posterior segment as TAT-aFGF could not be detected in the deeper layers of the cornea like the stroma and endothelium and were only found in the superficial corneal epithelium cells, meaning it was not fully able to penetrate and pass through the cornea. The mechanism of cellular uptake of TAT is expected to be cell specific because of selective uptake by RGCs.

Zhang et al. (2015) [23] demonstrated delivery of endostatin (Es), a specific inhibitor of endothelial cell proliferation and angiogenesis, with TAT via eye drop instillation, tested their ocular penetration system in Kunming mice, and evaluated their inhibitory effects in a choroidal neovascularization (CNV) model in 5-week-old male C57BL/6 mice by immunohistochemistry (IHC). Retinal distribution, as observed by IHC, revealed a good distribution profile of TAT-Es in the retina after eye drop application whereas retinas treated with unconjugated Es showed no detection. The results were supported by the inhibitory effects of TAT-Es on CNV areas observed in choroidal flatmounts. There was a significant reduction in CNV areas in mice treated with TAT-Es eye drops (1378.4 ± 154.3 μm^2^) compared with negative controls (2623.6 ± 240.9 μm^2^), while Es only eye drops showed no such protective effects and were similar to negative controls (2514.7 ± 260.7 μm^2^). They also observed that TAT-Es eye drops showed almost similar inhibitory effects as with the positive control group receiving intravitreal injections of Avastin for 2 weeks (863.5 ± 50.1 μm^2^) and intravitreally injected TAT-Es (961.2 ± 86.9 μm^2^), which means that eye drop treatment was as efficient as intravitreal injections and could easily replace it. There have been multiple mechanisms suggested in the study for cellular internalization including energy independent direct translocation for a small portion, but majorly endocytosis dominated by macropinocytosis and relatively smaller instances of clathrin and caveolae-mediated endocytosis. It was speculated that since TAT bears a high net positive charge and could easily interact with negatively charged proteoglycans on the cell membrane like heparan sulphate that work as receptors for positively charged molecules, it may initiate a receptor mediated uptake via macropinocytosis.

Li et al. (2016) [25] used TAT combined with a tripeptide of arginine-glycine-aspartic (RGD) as a novel fusion protein for cell-specific Es therapy for anti-angiogenesis effects and inhibition of retinal neovascularization (RNV) in DR in an oxygen-induced retinopathy (OIR) model in C57BL/6 mice via eye drops. ELISA detected 14-fold higher concentrations (9.42 ± 0.77 ng/mg) of TAT-Es-RGD in the retina at 1 h after topical application compared to that detected with unconjugated Es (0.68 ± 0.10 ng/mg). The concentration of TAT-Es-RGD was slightly higher than TAT-Es (8.43 ± 0.85 ng/mg) owing to the endothelial cell targeting ability imparted by RGD that binds preferentially to the α_v_β_3_ integrin expressed on angiogenic endothelial cells. Concentrations of TAT-Es and TAT-Es-RGD peaked at 1.00 ± 0.00 h and 1.125 ± 0.35 h, respectively, and their half-lives were 9.64 ± 1.31 h and 11.03 ± 1.95 h, respectively. Further, fluorescence microscopy analysis of retinal flat mounts of OIR to check the inhibitory effect of TAT-Es-RGD on RNV manifested a significant reduction in avascular areas and vessel tufts in the groups treated with TAT-Es and TAT-Es-RGD via eye drops compared with those of the negative control group. While the Es and Es-RGD treated groups were no different from the negative control group. Breaking of endothelial cells from the inner limiting membrane (ILM) of the retina was also quantified as an evaluation index for hyperoxia-induced neovascularization. A histological analysis of the ILM of retinas proved there was a significant decrease in the number of endothelial cell nuclei, which broke through the ILM in areas of neovascularization in the TAT-Es and TAT-Es-RGD groups as compared with negative controls, whereas Es and Es-RGD groups had no significant difference when compared with negative controls. In addition to these results, levels of VEGF that induced angiogenesis in OIR mice were detected using IHC. The levels of VEGF in TAT-Es and TAT-Es-RGD groups were much lower than that of the negative control groups, whereas in the Es and Es-RGD groups the levels were similar to that of the negative control group. Eye drop delivery of TAT-Es-RGD reduced VEGF expression by a similar amount as intravitreal injection, signifying how eye drops can easily replace invasive treatments.

Chu et al. (2017) [27] reported topical ocular delivery of PEG-PLGA nanoparticles (NP) modified with TAT-RGD dual peptides for penetration enhancement and targeting in laser induced CNV models in male Brown Norway rats. Choroid-sclera flatmounts of CNV regions were examined using confocal microscopy to determine the extent of penetration and distribution of NPs. Fluorescent intensity in the CNV region was maximum in the groups treated with RGD and TAT dual-modified NPs and was almost 11 times higher than in the groups treated with non-modified NPs, which showed negligible emission after eye drop administration in the CNV area. Fluorescence intensity in different treatment groups was measured in the order of RGD-TAT-NP > TAT-NP > RGD-NP > NP. TAT modified NPs were able to penetrate the ocular barriers and internalize into retina-choroid, however the internalization was scattered and not focused in the CNV sites due to the absence of selectivity in TAT. RGD and TAT dual-modified NPs exhibited maximum localization and accumulation surrounding and particularly within the CNV region. This is attributed to the complementary targeting property of RGD peptide by binding with integrin α_v_β_3_.

Atlasz et al. (2019) [30] studied TAT bound delivery of vasoactive intestinal peptide (VIP), a retinoprotective peptide, and pituitary adenylate cyclase activating polypeptide (PACAP), a potent anti-inflammatory peptide, in male rats via topical eye drop administration. Retinal penetration was evaluated by fluorescence microscopy. Retinas of mice treated with eye drops of PACAP-TAT and VIP-TAT showed much higher fluorescence density per unit area than in retinas treated with PACAP/VIP, indicating that PACAP-TAT/VIP-TAT reached the retina more efficiently than PACAP/VIP. The efficiency for traversing eye to retina (EtE) was also calculated. EtE for PACAP-TAT/VIP-TAT (3.66% ± 0.67%, 3.05% ± 0.58%) was about three times higher than that for PACAP/VIP (1.23% ± 0.56%, 0.97% ± 0.47%).

#### 3.1.2. Penetratin

Penetratin is another commonly used CPP derived from a nonviral antennapedia homeodomain protein in *Drosophila melanogaster*, with a general sequence of RQIKIWFQNRRMKWKKK and molecular weight 2.733 kDa. Liu et al. (2014) [22] first reported use of penetratin as a penetration enhancer for drug delivery to the fundus oculi via eye drop instillation. Penetration efficiency and biodistribution of penetratin was evaluated in 17-week-old male Sprague−Dawley rats using fluorescence microscopy. Within 10 min of administration, penetratin was able to internalize in both anterior and posterior segments of the eye. In the posterior segments it was distributed widely in almost all layers of retina-choroid with maximum uptake by the photoreceptor segments and retinal pigment epithelium (RPE) in the subretinal space displaying intense fluorescence in rods and cones, a moderate and uniform distribution in the GCL, inner plexiform layers (IPL), outer plexiform layer (OPL), and mild uptake into the inner nuclear layer (INL) and outer nuclear layer (ONL). Accumulation peaked at 30 min as evidenced by fluorescence intensity and was detectable for as long as 6 h after application suggesting lasting retention in these tissues. An in-vitro cytotoxicity assessment of penetratin by MTT assay was performed on human conjunctival epithelial cells (NHCs) and was found to be the least toxic among a group of CPPs, with a high IC_50_ value of approximately 2.5 mM, even higher than that of TAT (2 mM). However, even at a concentration as high as 30 mM of penetratin, the cell viability was still close to 100%. Both transcorneal and conjunctival-scleral transport pathways contribute to penetratin getting into the posterior segment and its penetration ability can be attributed to the high positive charge, its amphipathicity and polyproline type II (PPII) helix structure as observed by circular dichroism.

Liu et al. (2016) [26] continued their explorations with penetratin and this time around demonstrated retinal gene delivery of red fluorescent protein plasmid (pRFP) by PAMAM dendrimers conjugated with penetratin, in 17 week old male Sprague−Dawley rats via topical instillation. Their penetration and distribution were evaluated by fluorescence microscopy. Penetratin-PAMAM-pRFP complexes internalized into the posterior segment in merely 10 min, accumulation peaked from 1 to 2 h, and persisted until 8 h post administration. Maximum fluorescence intensity was observed in photoreceptors and RPE, followed by GCL, IPL, OPL, and the choroid, and faintish fluorescence in the INL and ONL. In contrast, eyes treated with nude pRFP had no detectable fluorescence in the retina. Furthermore, in-vivo gene expression was also measured and revealed that nude plasmids could not transfect retinas, while penetratin-PAMAM-pRFP complexes displayed strong transfections in photoreceptors, IPL and OPL, which was in-line with the results of distribution.

Jiang et al. (2017) [29] compared the in-vivo penetration ability of different derivatives of penetratin with that of wild-type penetratin via topical eye drop administration in male mice. Pharmacokinetics and biodistribution of hydrophobic and hydrophilic derivatives in comparison with wild-type penetratin were estimated by fluorescence microscopy. Initially all of the peptides were able to internalize in both anterior and posterior segments within 10 min as evident by bright fluorescence and their concentrations peaked around 1-h post eye drop instillation. However, the retinas treated with the hydrophobic derivatives emitted overwhelming levels of fluorescence, revealing that the hydrophobic derivatives internalized more efficiently than did the wild-type penetratin. In contrast, the retinas treated with the hydrophilic derivatives showed weaker fluorescence compared with wild-type penetratin implying impaired penetration ability. To back the results, it has been suggested that higher hydrophobicity induces more helix content in the peptide structure as seen in circular dichroism studies, which in turn helps in stronger affinity interactions with the biological membranes thus facilitating higher permeability. Toxicity analyses performed in-vitro on human corneal and conjunctival epithelial cells via MTT assay revealed no cytotoxicity, and ex-vivo in excised rabbit corneas and sclera by measuring the hydration values also showed good biocompatibility and no leakage during translocation.

Yang et al. (2019) [32] studied penetratin and RGD peptide modified nanocarriers (NCs) of PEG-PAMAM dendrimers for enhanced penetration and complementary targeting to the ocular posterior segment in male ICR mice via topical application. Ocular permeability and posterior segment distribution as estimated by fluorescence microscopy exhibited that although all the NCs were able to penetrate the posterior segment, the mouse retinas treated with NCs conjugated with penetratin or RGD-penetratin showed almost 3 times stronger fluorescence intensity than the retinas treated with non-conjugated nanocarriers, at their peak concentrations at 8 h post application. Penetratin-RGD-NCs initially entered the retinas within 1 h, penetrated the RPE within 2 h, and then gradually internalized to the inner retinal layers, and the accumulation reached a maximum at 8 h post administration. In addition, these retinas kept emitting measurable fluorescence for up to 12 h and were still detectable 24 h after instillation, indicating that the penetratin-RGD modified NCs resided in the retinas persistently, which was not the case with retinas treated with non-modified NCs as their fluorescence died out within 12 h. MTT viability assays to establish in-vitro cytotoxicity of the peptide-NC complexes in human conjunctival epithelial cells (NHCs), human corneal epithelial cells (HCECs), and human umbilical vein endothelial cells (HUVECs) revealed that these complexes had no detectable cytotoxicity, and the percentage cell survival was 90–110% after incubation with 0.2–20 µM NCs. PEG conjugation significantly attenuated the cytotoxicity of PAMAM dendrimers because of reduced amino groups on the surface and surface passivation, thus preventing contact between cells and primary amines.

#### 3.1.3. Other CPPs

Johnson et al. (2008) [20] reported a novel CPP called peptide for ocular delivery (POD) with a general sequence GGG(ARKKAAKA)_4_ and molecular weight 3.5 kDa that could efficiently reach and deliver cargo to the fundus oculi in-vivo in C57BL/6J mice by topical eye drop instillation. Pharmacokinetics and ocular distribution in the posterior segment were determined by microscopic analysis, which revealed that in the eyes treated with POD–drug complexes the sclera/choroid and dura of the optic nerve were transduced by these complexes in 45 min after application and it was eliminated within 24 h, whereas no such staining was observed in the groups with non-conjugated drug. They have also suggested that cellular uptake of POD is temperature dependant and does not happen by endocytosis as no reduction in peptide uptake was observed in-vitro in the presence of endocytosis inhibitors. Instead, a direct entry mechanism without plasma membrane fixation is supported as also cell-surface proteoglycans inhibited the uptake of POD in-vitro.

de Cogan et al. (2017) [28] tested delivery of anti-VEGF drugs using another novel poly-arginine based CPP: (5[6]-carboxyfluorescein-RRRRRR-COOH) for their penetration and pharmacokinetics in male Sprague–Dawley rats, and inhibitory effects in a CNV model in wild-type (WT) C57BL/6J mice, via eye drop application. Penetration and biodistribution was evaluated by ELISA. Mice treated with CPP–drug conjugates showed a peak drug concentration about 10–11 times higher in contrast to the groups treated with non-conjugated anti-VEGF drugs. The levels of drug accumulation in CPP–drug conjugates in the retina peaked at 40 min after topical administration representing an upwards of 0.2% of the applied pay load, followed by elimination in 24 h. To estimate the inhibitory effects of CPP–drug conjugates on CNV lesions, fluorescein angiography (FA), infrared imaging, and IHC of disease biomarkers (collagen IV and isolectin B4) in RPE/choroid flatmounts showed that CPP–drug complexes caused considerable reduction to two-thirds extent in the area of scar and neovascularization, whereas CNVs treated with non-complexed drugs had no apparent differences from the untreated or negative control groups. Additionally, the inhibitory effects of eye drop instilled CPP–drug conjugates were similar to those achieved by intravitreal injection of drugs or CPP–drug conjugates, implying that non-invasive topical administration may be a preferred mode for treating posterior segment diseases. No cytotoxicity was detected for CPPs in in-vitro MTT assays in rat retinal cultures, human ARPE-19 cells, and human corneal fibroblasts.

### 3.2. Chitosan

Chitosan is the second most abundant natural polysaccharide derived from deacetylation of chitin and has a sugar backbone of β-1,4-linked glucosamine. It is a polycationic biopolymer and a structural component of the exoskeletons of crustaceans and insects, but also found in some fungi. It bears many favourable properties, like non-cytotoxicity, biocompatibility, and biodegradability and has been used in pharmaceutical and biomedical fields for decades, playing a tremendous role in advancement of drug delivery systems [36]. Chitosan and its derivatives has been long used as a penetration enhancer for transdermal, oral, nasal, buccal, and vaginal drug delivery [37]. In ocular drug delivery it finds wide use as an excipient as penetration enhancer owing to its highly mucoadhesive nature that allows prolonging drug residence in the precorneal area and thus resulting in higher bioavailability.

Li et al. (2019) [31] used chitosan as a penetration enhancer on liposomes for ocular delivery of triamcinolone acetonide via eye drop instillation to the posterior segment of the eye in C57BL/6 mice. Optical coherence tomography (OCT) signal intensities to evaluate penetration and distribution in the eyes showed that eyes treated with chitosan coated liposomes emitted relatively higher levels of signal from the posterior segment of the eye when compared with non-coated liposome treated eyes, though it was not significantly different. The higher levels of the relative signal intensity in the posterior segment of the eye started at 10 min and peaked at 6 h after topical administration. Higher signal intensities from eyes treated with chitosan coated liposomes lasted longer (12 h) than that of non-chitosan groups (10 h) indicating that apart from slightly higher penetration, chitosan also enabled longer retention in the posterior segment.

Wang et al. (2020) [33] demonstrated the penetration enhancing ability of carboxymethyl chitosan (CMCS) coating over nanocomposites to deliver dexamethasone (DEX) to the posterior segment of the eye in New Zealand albino rabbits through topical eye drops. In-vivo activity and distribution of the CMCS coated nanocomposites was estimated by measuring the DEX concentrations in retina-choroid tissue homogenates using HPLC. Eyes treated with CMCS conjugated nanocomposites showed the highest DEX concentration in retina-choroid as compared to the groups treated with commercially available DEX eye drops and non-CMCS nanocarriers. In the commercial eye drop groups, DEX was not detectable in the tissues at any time point and its concentration was below detection limits. Even with the non-CMCS nanocarriers, DEX was cleared from the choroid-retina within 1 h post application suggesting impaired retention of the nanocomposites. At 30 min however, DEX tissue concentration in CMCS-nanocomposite groups was 2.5 times higher than that in the non-CMCS nanocomposite groups, which peaked at 1 h and was still detectable up to 3 h post administration, implying greater penetration and longer residence time in retina-choroid. The in-vivo transport pathway for internalization of CMCS was visualized using fluorescence imaging of rabbit eyes after eye drop instillation which measured the fluorescence intensity of all ocular tissues to track the CMCS coated nanocomposites revealed an overwhelmingly stronger fluorescence in the conjunctival-scleral pathway than in the corneal pathway, indicating that CMCS is transported to the posterior segment via the conjunctival-scleral route. The primary mechanism of CMCS uptake in the human conjunctival epithelial cells is via energy-dependent and clathrin-mediated endocytosis as evidenced in-vitro by about 54% reduction in cellular uptake when inhibitors of clathrin-mediated endocytosis were used.

### 3.3. Benzalkonium Chloride (BAC)

BAC is an age-old antimicrobial preservative widely used in ophthalmic formulations and may act as a penetration enhancer, generally to the anterior segment tissues. It facilitates penetration by breaking down the physiological and anatomical diffusion barriers for solute and solvent molecules located in the outer layer of the epithelium.

Mahaling and Katti (2016) [24] used BAC for enhanced penetration of nanoparticles into ocular tissues via topical eye drop administration in pigmented mice. Bioactivity and distribution of BAC coated nanoparticles was determined by fluorescence microscopy. Although the effect of BAC on ocular penetration of nanoparticles and localisation in the retina showed promising results, the enhancement effect was not very pronounced. However, there was some vaguely reported significant increase in bioavailability of BAC coated nanoparticles in the retina-choroid-sclera, as compared to non-coated nanoparticles, which also varied depending on the surface properties of nanoparticles.

### 3.4. Risk of Bias (RoB)

RoB analysis carried out on the methodological quality of the 14 included studies is depicted as a stacked bar-chart in Figure 2. In the included studies, there is a high risk of bias since none of the studies reported treatment concealment, random housing of animals, blinding of interventions, nor assessed outcomes randomly. More than 55% of studies did not include information on random sequence generation for animals, specified the correct timing of randomisation, nor provided sample size calculations. Only 70% of the studies described baseline characteristics, 85% specified primary outcomes but 100% were deemed free from contamination (i.e., when control groups inadvertently receive the treatment or are exposed to the treatment). The RoB analysis of the 14 articles demonstrates that some of the animal studies possess a high degree of bias that may negate/question the findings of those studies.

## 4. Discussion

We systematically reviewed all the available literature on experimental PEs for topical delivery to the posterior segment of the eye. From the primary search result of 6540 articles, we filtered the relevant publications to a final 14 in conformity with our inclusion criteria and qualitatively analysed them since a meta-analysis was not possible due to lack of a common standardized quantitative measurement of penetration efficacy in the selected studies. The search results included all the different ocular PEs that have been reported, however most of them were only effective for anterior segment delivery. For decades, PEs have been used in topical transdermal drug delivery and by oral delivery to the intestinal mucosa, and it is only recently that they are being employed for ocular delivery. Topical eye drop formulations did not evolve functionally for diseases of the fundus oculi, but now there seems to be a renewed vigour in their development especially when PEs can overcome the biggest hinderance of low uptake and bioavailability in the posterior segment tissues.

This systematic review revealed that CPPs are the most favourable PEs for drug delivery to the fundus oculi with high cellular internalization and broad biodistribution in retina-choroid. CPPs have demonstrated their ability to increase the in-vivo uptake of various cargoes including protein drugs [21,23,25,28], nanoparticles [27], genes and nucleic acids [26], dendrimers [26,32], peptides [30], etc., across the ocular biobarriers and into the posterior segment via eye drop instillation. In some cases, topical PEs have affected a similar uptake and biodistribution as observed with intravitreal or intraocular injections thus providing a safe and effective alternative to invasive treatment procedures [23,25,28]. Pharmacokinetic distribution of all the CPPs are very impressive and they were detected in almost every layer of the retina-choroid like RGCs–GCL, photoreceptor segments, RPE, rods and cones, IPL, OPL, ILM, INL, and ONL. Although their penetration mechanisms and transport pathways in the eye need more detailed investigations, this review does show how they may follow all the different internalization mechanisms from passive direct translocation across cellular membranes [20,23] to active endocytosis including proteoglycan-mediated macropinocytosis and clathrin or caveolae-mediated endocytosis [23]. All of these pathways are consistent with their general cellular uptake [38,39] and hence, can access the ocular posterior segment from both transcorneal and conjunctival-scleral pathways [22]. CPP studies have also encompassed almost all of the different animal models of major posterior segment pathophysiology including IR injury [21], CNV [23,27,28], RNV, and OIR [25]. Interestingly the CPPs could also be modified with RGD tripeptide for cell-specific and targeted uptake owing to its binding affinity to the α_v_β_3_ integrin expressed on angiogenic endothelial cells [25,27,32]. Additionally, it is worth noting their negligible or non-cytotoxic nature as demonstrated by in-vitro cell viability assays in different cell lines including human corneal epithelial cells (HCECs) and human conjunctival epithelial cells (HConEpic) [22,29,32], rat retinal cultures, human ARPE-19 cells, and human corneal fibroblasts [28].

BAC is a well-known antimicrobial preservative that has been used in ophthalmic eye drop formulations for a long time. Here we see it does work on penetration into the ocular posterior segment but only to a very limited and negligible extent and thus at best can be used for treatment of anterior segment diseases also by virtue of its internationalization mechanism as it works by breaking down the tight junctions of surface epithelia on cornea [24]. Chitosan is reported to enhance the precorneal retention time of drugs due to its mucoadhesive nature thus effectively preventing elimination by tear flow. Chitosan was used as a PE for posterior segment uptake without any encouraging efficacy but could enhance the residence of liposomes in fundus oculi to some extent [31], so we drew the inference that it could be used in combination with CPPs as to increase the residence time of eye drop at the corneal surface resulting in higher uptake and will also increase therapeutic efficacy of the drug by increased retention in the target tissues. CMCS on the other hand showed very promising results with a 2.5 times enhancement in the cargo uptake into the retina-choroid though their major transport was through conjunctival-scleral pathway [33]. As opposed to chitosan, CMCS is anionic in nature and bears a net negative charge but still retains mucoadhesive properties of chitosan residues and enables an increased precorneal and posterior segment residence time, and its uptake is defined as majorly by clathrin-mediated endocytosis.

The risk of bias analysis on the methodologies of selected studies showed a relatively high tendency of bias. Though more than >70% of the studies reported baseline characteristics of animals, a poor randomization of allocation to experimental and control groups and absolutely no concealment of this gives rise to high selection bias. None of the studies report random housing of animals and blinding of investigators and animal caregivers at the intervention stage, thus resulting in extremely high performance bias. Random housing of animals before and after intervention is important because housing conditions like lighting, humidity, temperature, etc., can influence outcomes of the study and behaviour of animals and thus should be comparable in experimental groups [18]. While assessing the outcome, none of the studies report selecting animals at random for analysis and more than >90% of the studies did not ensure blinding of the investigator who made the assessment, which results in very high detection bias. A moderate attrition bias is observed in more studies addressing the incomplete outcome data, however with insufficient information on the methodology used. A good >85% of studies have specified primary outcomes for each intervention, however more than >57% did not calculate a sample size and the few that did calculate had insufficient information on methodology in terms of reporting bias. On the brighter side, there is absolutely no risk of contamination in any of the studies, i.e., neither did the experimental groups receive any additional treatments besides the primary intervention nor were the controls exposed to inadvertent interventions. A 50:50 bias due to unit of analysis errors is reported where one eye of the same animal was used as experimental and control. The high risk of bias can be avoided by adhering to a set of standardised techniques in animal experiments based on the ARRIVE guidelines (Animal Research: Reporting of In Vivo Experiments) [40,41].

### Limitations

The leading limitation of this article was that a meta-analysis was not possible due to the lack of quantitative data in the included studies. Another limitation is that almost all of the included studies used IHC to assess penetration of drugs to the posterior segment, which is subjective and is prone to inconsistencies due to antibody characteristics such as relative affinity, specificity, and sensitivity to detect the target antigen. The use of poorly characterised antibodies has given rise to the inability of researchers to replicate published data, especially in the analysis of low abundance proteins such as G protein-coupled receptors, steroid hormones, ion channels, and transporters [42]. A quantitative method such as ELISA or HPLC would be preferable when working out posterior segment PE delivery efficiency. Therefore, future studies should take into account these limitations when designing studies to explore the efficacy of PEs to deliver their payload to the posterior segment of the eye.

## 5. Conclusions

This systematic review suggests that cationic and amphipathic CPPs like TAT and penetratin are the most promising PE candidates in the present domain for facilitating an increased passage of cargo across the defensive barriers of the eye and into the fundus oculi via topical eye drop administration. These CPPs not only exhibit greater permeability at ocular barriers but also an efficient biodistribution and cellular uptake in the tissues of the ocular posterior segment. A cell-specific and targeted uptake in diseased sites can be achieved in combination with RGD tripeptide. More in-vivo studies are required however, to determine their quantitative efficacy in terms of the proportional bioavailability in the posterior segments of the eye with that of topically applied payload. Most of the in-vivo uptake into the retina-choroid takes place via the conjunctival-scleral pathway. Bioavailability in these tissues can be further enhanced if CPPs can be used in combination with mucoadhesive polymers to increase precorneal residence time and thus effectively allowing for an augmented transcorneal internalization. Further investigations should be carried out on their in-vivo penetration mechanisms and toxicity and also whether the penetration causes any irreversible alterations in barrier properties of the defensive barriers, which can cause more harm than good.

## Figures and Tables

**Figure 1 pharmaceutics-13-00276-f001:**
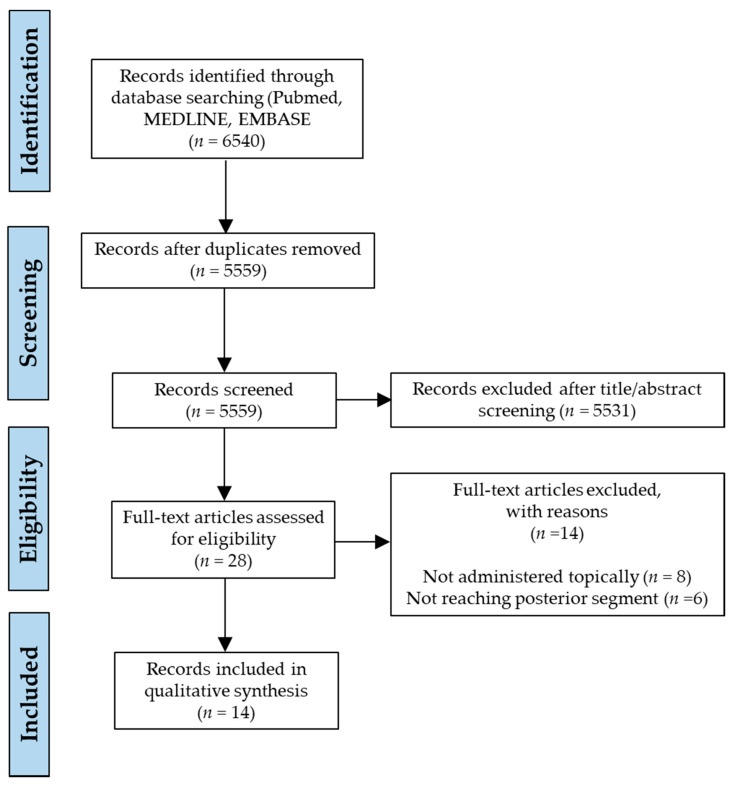
Preferred Reporting Items for Systematic Reviews and Meta-Analyses (PRISMA) flow diagram demonstrating the literature search strategy.

**Figure 2 pharmaceutics-13-00276-f002:**
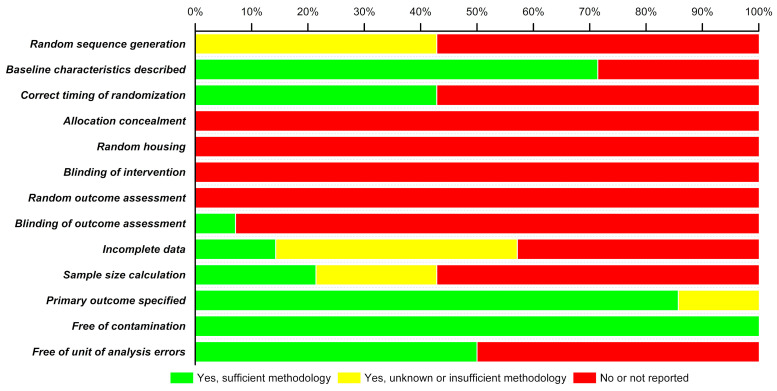
Risk of bias (RoB) analysis.

**Table 1 pharmaceutics-13-00276-t001:** Study characteristics.

Publication	Penetration Enhancers (PE)	Animal Model	In-Vivo Efficacy and Biodistribution
Johnson et al., 2008 [20]	CPP: Peptide for Ocular Delivery (POD)—GGG(ARKKAAKA)_4_	C57BL/6J mice	Microscopy—POD conjugates transduced sclera/choroid and dura of the optic nerve 45 min post application and was eliminated within 24 h; no staining in non-POD groups
Wang et al., 2010 [21]	TAT—CPP	Sprague–Dawley rats	Immunohistochemistry—TAT conjugates readily detected in retina 30 min post administration, mainly in RGCs in the GCL, peaked around 30 min to 1 h, detectable until up-to 8 h; no detection in non-TAT retinas
Liu et al., 2014 [22]	Penetratin—CPP	Sprague–Dawley rats	Fluorescence microscopy—Penetratin reached both anterior and posterior segments of the eye 10 min after application, fluorescence detected in GCL, IPL, OPL, INL, ONL, RPE, and the photoreceptor segments; peaked at 30 min and persisted for 6 h; no fluorescence detected in untreated eyes
Zhang et al., 2015 [23]	TAT—CPP	Kunming mice and C57BL/6 mice	Immunohistochemistry—Groups treated with TAT conjugated drug had high drug distribution in the retina after eye drop administration; unconjugated drug groups showed no drug
Mahaling and Katti, 2016 [24]	Benzalkonium chloride (BAC)	Pigmented mice	Fluorescence microscopy—Results indicated that presence of BAC led to varying degrees of increase in bioavailability of different nanoparticles to the retina as compared to non-coated nanoparticles
Li et al., 2016 [25]	TAT—CPP	C57BL/6 mice	Quantitative analysis (ELISA)—Concentration of drug in the retina groups treated with TAT conjugated drug was approximately 14 times higher than in those with non-conjugated drug; concentration peaked around 1 h, and the half-life was around 11 h
Liu et al., 2016 [26]	Penetratin—CPP	Sprague–Dawley rats	Fluorescence microscopy—Penetratin conjugated complexes reached the posterior segment in 10 min post application, fluorescence detected in photoreceptor segments, RPE, GCL, IPL, OPL, choroid, INL, and ONL; no fluorescence detected in eyes treated with unconjugated complexes
Chu et al., 2017 [27]	TAT—CPP	Brown Norway rats	Confocal microscopy—Eyes treated with TAT modified complexes exhibited 11 times higher fluorescence in the retina-choroid than those treated with non-modified complexes.
de Cogan et al., 2017 [28]	CPP: (5[6]-carboxyfluorescein-RRRRRR-COOH)	Adult Sprague- Dawley rats and Wild-type (WT) C57BL/6J mice	Quantitative analysis (ELISA)—Concentration of drug detected in groups treated with CPP conjugated drug was 10–11 times higher compared to groups treated with non-conjugated drug; drug levels peaked at 40 min post administration and was eliminated by 24 h
Jiang et al., 2017 [29]	Penetratin derivatives—CPP	Mice	Fluorescence microscopy—Fluorescence from retinas treated with hydrophobic penetratin derivatives > wild-type penetratin > hydrophilic derivatives; peptides reached the posterior segment within 10 min post instillation and fluorescence peaked around 1 h
Atlasz et al., 2019 [30]	TAT—CPP	Rats	Fluorescence microscopy—TAT bound complexes reached the retina with an efficiency of about three-fold higher than that of non-conjugated drugs
Li et al., 2019 [31]	Chitosan	C57BL/6 mice	Optical coherence tomography—Eyes treated with chitosan conjugates emitted higher levels of fluorescence from the posterior segment than those eyes treated without chitosan; fluorescence detection started at 10 min, peaked at 6 h, and disappeared 12 h post application
Yang et al., 2019 [32]	Penetratin—CPP	ICR mice	Fluorescence microscopy—Retinas treated with penetratin modified complexes demonstrated much higher fluorescence intensities (~3 times greater) as compared to retinas with non-modified complexes; detected in retina 1 h post administration, in RPE within 2 h, peaked at 8 h and still detectable until 12 h
Wang et al., 2020 [33]	Carboxymethyl chitosan (CMCS)	New Zealand albino rabbits	Quantitative analysis (HPLC)—Drug concentration in retina-choroid treated with CMCS conjugates were 2.5 times higher than in those with non-conjugated drugs; drug concentration peaked at 1 h and was detectable up-to 3 h post application

## Data Availability

All data generated as part of this study are included in the article.

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
