# Peer review of "Penetration Enhancers for Topical Drug Delivery to the Ocular Posterior Segment—A Systematic Review"

_pharmaceutics, 2021, doi:10.3390/pharmaceutics13020276_

Round 1
Reviewer 1 Report
This interesting review “Penetration enhancers for topical drug delivery to the ocular posterior segment – A systematic revie” addresses an overview of the most relevant manuscripts reported on ocular penetration enhancers, focusing on penetration enhancers and among these, the most efficient at delivering drugs to the posterior segment of the eye, when topically applied, were selected.
The subject is very interesting, the eye diseases and the involved health, social and economic problems are well presented, the used Inclusion and Exclusion Criteria are valuable and the previously studies are well described. I suggest the publication in “Pharmaceutics” without revisions.
Author Response
Thank you for your comments and your recommendation of publishing without any changes.
Reviewer 2 Report
The authors bring with this systematic review into the discussion the penetration enhancers for eye-drop formulations for a non-invasive topical administration to efficiently treat the posterior ocular segment diseases.
The manuscript has a good flow and the design of the metadata analysis respects the requirements.
My suggestions and observations are as follows:
- Please , improve the resolution of the Flow diagram for PRISMA (figure 1).
- The list of the references seems to be poor in entries. Even working majority, or only with large data bases, the list could be improved.
- Please, correct the references list according to the Journal Indications for the Authors.
Author Response
Comment: Please , improve the resolution of the Flow diagram for PRISMA (figure 1).
Author Response: We have included an improved resolution PRISMA flow chart (Fig. 1).
Comment: The list of the references seems to be poor in entries. Even working majority, or only with large data bases, the list could be improved.
Author response: We have improved the references and checked them for consistency.
Comment: Please, correct the references list according to the Journal Indications for the Authors.
Author response: We have followed the Journal guidelines and the references should now be in the correct format.
Reviewer 3 Report
Delivery of therapeutics to back of the eye is a major challenge and in-depth literature review on this topic will aid other researchers to dive deep into domains that is of importance. In this review title "Penetration enhancers for topical drug delivery to the ocular posterior segment", authors have done a great job distilling key ideas and convincing the readers that those are the domains worth considering. I personally think this is a timely review done with robust methodology and worth publishing.
Author Response
Thank you for your kind comments. There were no comments that required author responses.